# Synthesis and Biological Evaluation of Novel Allobetulon/Allobetulin–Nucleoside Conjugates as AntitumorAgents

**DOI:** 10.3390/molecules27154738

**Published:** 2022-07-25

**Authors:** Yanli Wang, Xiaowan Huang, Xiao Zhang, Jingchen Wang, Keyan Li, Guotao Liu, Kexin Lu, Xiang Zhang, Chengping Xie, Teresa Zheng, Yung-Yi Cheng, Qiang Wang

**Affiliations:** 1School of Medicine, Huanghe Science and Technology College, Zhengzhou 450063, China; hblyfe@163.com; 2National Health Commission Key Laboratory of Birth Defect Prevention, Henan Institute of Reproductive Health Science and Technology, Zhengzhou 450002, China; lky130277680782022@163.com (K.L.); liuguotao33@163.com (G.L.); 3High & New Technology Research Center, Henan Academy of Science, Zhengzhou 450002, China; 18336708203@163.com (X.H.); zxzsy0405@163.com (X.Z.); xiechp@ceprei.com (C.X.); 4BGI College & Henan Institute of Medical and Pharmaceutical Science, Zhengzhou University, Zhengzhou 450052, China; zhangxiao7011@163.com (X.Z.); chen13021083099@163.com (J.W.); kexin14590@163.com (K.L.); 5Division of Pharmacoengineering and Molecular Pharmaceutics, Eshelman School of Pharmacy, The University of North Carolina at Chapel Hill, Chapel Hill, NC 27599-7568, USA; tzheng1999@gmail.com

**Keywords:** pentacyclic triterpene, allobetulin, conjugates, antitumor activity, apoptosis, nucleosides

## Abstract

Allobetulin is structurally similar tobetulinic acid, inducing the apoptosis of cancer cells with low toxicity. However, both of them exhibited weak antiproliferation against several tumor cell lines. Therefore, the new series of allobetulon/allobetulin–nucleoside conjugates **9a**–**10i** were designed and synthesized for potency improvement. Compounds **9b**, **9e**, **10a,** and **10d** showed promising antiproliferative activity toward six tested cell lines, compared to zidovudine, cisplatin, and oxaliplatin based on their antitumor activity results. Among them, compound **10d** exhibited much more potent antiproliferative activity against SMMC-7721, HepG2, MNK-45, SW620, and A549 human cancer cell lines than cisplatin and oxaliplatin. In the preliminary study for the mechanism of action, compound **10d** induced cell apoptosis and autophagy in SMMC cells, resulting in antiproliferation and G0/G1 cell cycle arrest by regulating protein expression levels of Bax, Bcl-2, and LC3. Consequently, the nucleoside-conjugated allobetulin (**10d**) evidenced that nucleoside substitution was a viable strategy to improve allobetulin/allobetulon’s antitumor activity based on our present study.

## 1. Introduction

In the population of people under the age of 70, cancer is one of the leading causes of death in six-tenths of the countries in the world, according to the World Health Organization (WHO) report in 2019 [1]. Globally, over 10 million new cases and deaths occurred in 2020 [2]. Chemotherapy is usedas a first-line anticancer remedy due to this treatment’s efficacy, despite its significant adverse effects [3,4,5]. Seeking alternative medicines with acceptable or less adverse effects and promising anticancer activities are the goals of anticancer drug discovery and development. Natural products are considered sources for drug discovery, and the discovery of such products leads to promising clinical outcomes.

Pentacyclic triterpenes (PTs) from natural products, including betulin (**1**), betulinic acid (**2**), oleanolic acid (**3**), and ursolic acid (**4**) (Figure 1), have attracted much attention because of their various biological activities (e.g.,antiviral, antineoplastic, antiparasitic, antibacterial, anti-inflammatory, antiulcer, antifeedant, antidiabetic, anticarcinogenic, hepatoprotective, nephroprotective, neuroprotective, and cardioprotective activities) [6,7,8,9,10,11,12,13,14,15,16]. Their multi-target behaviorin cancer allows them to bein the forefrontofa new generation of anticancer drug candidates [17]. Betulinic acid (**2**), for instance, improved reactive oxygen species production, triggered mitochondrial-mediated apoptosis via the caspase-dependent signaling pathway, and was linked to the p38 and stress-activated protein (SAP) kinase/c-jun*N*-terminal kinase (JNK) inseveral human cancer cell lines [18,19]. Allobetulin (19β,28-epoxyolenan-3-ol, **5**), one extractive substance from birch bark, is structurally similar to oleanolic acid (**3**). Regarding the configuration of H-18, the hydrogen atom at C-18 of allobetulin is in α-configuration rather than β-configurationin oleanane-type triterpenoids [20]. Allobetulin (**5**) is converted from the rearrangement reaction of botulin [21,22,23]. Therefore, allobetulin (**5**) not only belongs to the oleanane terpenoidsbut also as a “re-arranged”betulin derivative [24]. Additionally, the bioactivities of allobetulin (**5**) werereportedlyantiviral [25,26], anticancer [27,28,29,30,31,32], anti-inflammatory [33], antichlamydial [34], antioxidant [35,36], and neuroprotective effects [24].

Despite allobetulin (**5**) exhibiting multi-bioactivity, thestrength of its antiproliferation against several tumor cell lines is insufficient at micromolar concentrations. In our previous study, anti-HIV activities of betulinic acid derivatives were increased by conjugating them with nucleosides [37]. We hypothesized that various nucleoside pharmacophores introduced into allobetulin/allobetulon via click chemistry might also improve their potency. In the present study, we designed and synthesized allobetulin/allobetulon-nucleoside conjugates, and then estimated their antineoplastic activity. Subsequently, we investigated the mechanism of action for the promising candidate.

## 2. Results and Discussion

### 2.1. Chemistry

The synthesis of the 2-propargyl substituted intermediates isshown in Figure 1. Allobetulin (**5**) was obtained by Wagner–Meerwein rearrangement from betulin (**1**) in the presence of *p*-toluenesulfonic acid [24]. By Jones’ oxidation [30,37], allobetulin (**5**) reacted with CrO_3_ to produce allobetulon (**6**). The key intermediate (**7**) was obtained by the propargyl α-alkylation of allobetulon (**6**) reacted intheKN(SiMe_3_)_2_/Et_3_B system. Reduced 2α-propargyl-allobetulon was reacted with NaBH_4_ in isopropanol to preferentially produceanother intermediate, 2α-propargyl-allobetulin (**8**). Regarding the structural establishment, the NOE effect (Appendix A) between H-3 and H-23 indicated an equatorial position (β-orientation) for the OH group, and the NOE effects between H-2 and H-24/H-25 suggested an axial position (β-orientation) for H-2, and thus, an equatorial position (α-orientation) for the propargyl group. Furthermore, the spin–spin coupling constant (^3^*J*_H(2),H(3)_= 10.6 Hz, CDCl_3_) between H-3 and H-2 in the ^1^H-NMR spectrum of **8** (Appendix A) was consistent with axial positions for H-2 and H-3. The axial position (β-orientation) ofH-2 was also demonstrated by the X-ray diffraction determination of single crystals of compound **9c** (Figure 2).The 2-propargyl allobetulon (**7**) and allobetulin (**8**) were coupled with different azides (4′-azido-2′-deoxy-2′-fluoro-β-d-arabinocytidine (AFC), 4′-azido-2′-deoxy-2′-fluoro-β-d-arabinouridine (AFU), 4′-azido-β-d-ribocytidine (AZC), 4′-azido-β-d-ribouridine (AZU), 4′-azido-2′-deoxy-β-d-ribocytidine (AdC), 4′-azido-2′-deoxy-β-d-ribouridine (AdU), and AZT) via click chemistry to produce the target compounds **9a–i** and **10a–i**, respectively (Figure 2).

Finally, all the target compounds were fully characterized by ^1^H- and ^13^C-NMR, and HRMS spectra which were listed in the Appendix A). Additionally, the purity of the target compounds (≥95%) was confirmed by HPLC.

### 2.2. Biological Evaluation

#### 2.2.1. Antiproliferative Activities and Structure-Activity Relationship

Synthesized allobetulon/allobetulin–nucleoside derivatives were evaluated for their antitumor activity against six human tumor cell lines by Cell Counting Kit-8 (CCK8) assay, including a human hepatoma cell line (SMMC-7721), human hepatocellular carcinoma cell line (HepG2), human gastric cancer cell line (MNK-45), human non-small cell lung cell line (A549), human colorectal cell line (SW620), and human breast cancer cell line (MCF-7). Cisplatinand oxaliplatin, belonging to the platinum-based antineoplastic chemotherapy drugs on the World Health Organization’s List of Essential Medicines, interfere with DNA replication by binding to DNA [38]. In the CCK8 assay, cisplatin and oxaliplatin were used as the positive control. Additionally, zidovudine (azidothymidine, AZT), a kind of nucleoside analog reverse-transcriptase inhibitor (NRTI), was used as a positive control in this study.As a result, the 2-propargyl allobetulon (**7**) and allobetulin (**8**) (the synthetic scaffolds for other nucleoside–allobetulin/allobetulin conjugates) exhibited weak activity against six tested cell lines (Table 1). Among these derivatives, compound **9b** exhibited similar potency tooxaliplatin against MCF-7 cell line. Compared to cisplatin and oxaliplatin, compounds **9e**, **10a**, and **10d** showed significant potency against MNK-45 and SW620 cell lines. Interestingly, compound **10d** exhibited the lowest IC_50_ value for SMMC-7721 (5.57 μM), HepG2 (7.49 μM), MNK-45 (6.31 μM), SW620 (6.00 μM), and A549 (5.79 μM) cell lines. Allobetulon (**7**) exhibited lower potency than allobetulin (**8**) against SMMC-7721, HepG2, and A549 cell lines. Compared to the antineoplastic activities of zidovudine (>100 μM), the synthesized allobetulon/allobetulin–nucleoside derivatives had much more promising potency. Taken together, introducing various nucleosides to the scaffolds (**7** and **8**) could improve the antiproliferative activity against the tested cell lines. Conjugated nucleoside-substituted with fluorine glycosyl compounds (**9b**, **10a**, **10d**) presented promising antitumor activity. Consequently, compound **10d** exhibitedthe most promising antitumor activity against tested human cancer cell lines.

#### 2.2.2. Effects of Compound **10d** on Apoptosis, Autophagy, and Cell Cycle Study of SMMC-7721 Human Cancer Cells

Given the promising antineoplastic activities of compound **10d** against five tested human cancer cell lines, flow cytometry investigated the cell cycle distribution to determine whether compound **10d** influenced cell cycle progression. SMMC-7721 cells were exposed to the five different concentrations of compound **10d** (0, 1, 5, 10, 15 µM) and then subjected to flow cytometry. Cell cycle analysis showed increased accumulation of cells in the G0/G1 phase after treatment with compound **10d** (Figure 3B,C). Therefore, induction of G0/G1 cell cycle arrest in SMMC-7721 cell lines implied compound **10d** reduced cell proliferation by induction of G0/G1 cell cycle arrest. Some factors can trigger the G0/G1 cell cycle arrest, including apoptosis, cyclin-dependent kinase inhibition, the regulation of tumor suppressors, and autophagy. The apoptotic effect of compound **10d** toward SMMC-7721 was assessed by annexin VFITC and propidium iodide (PI) staining. SMMC-7721 was treated with dose-dependent concentrations of compound **10d** for 48 h and then subjected to flow cytometry analysis. As illustrated in Figure 3A, the percentage of the total proportion of apoptotic cells increased from the base value (control, 0.35%) to 46.41% for 15 µM, implying that compound **10d** induced apoptosis of SMMC-7721 cells. To clarify the potential factors for that, we investigated the critical regulators of cell apoptosis by Western blot analysis. Pro-apoptotic protein Bax and the anti-apoptotic protein Bcl-2 are well-known factors linked to the regulation of apoptosis. SMMC-7721 cells were treated with **10d** (1, 5, 10, 15 μM) for 48 h, and we then examined the associated protein levels (Bax, Bcl-2, and GAPDH) by Western blotting. As shown in Figure 3D, the expression of the pro-apoptotic protein Bax was upregulated, and that of the anti-apoptotic protein Bcl-2 was significantly down-regulated, both in a dose-dependent manner. The balance of Bax/Bcl-2 ratio is important in determining whether cells will undergo apoptosis. The ratio of Bax to Bcl-2 was dose-dependently increased in the range of 1 to 10 µM (Figure 3E). Based on these results, compound **10d** might lead to G0/G1 phase arrest in SMMC-7721 cells through apoptosis by Bax and Bcl-1 regulation. Additionally, G0/G1 phase arrest may be induced by autophagy. Therefore, the autophagy marker, LC3, was examined by Western blotting. Cells were treated with **10d** (1, 5, 10, 15 μM) for 48 h, and then the expression of LC3 was measured. As shown in Figure 3F, compound **10d** significantly induced the LC3 expression in a dose-dependent manner. Taken together, compound **10d** dose-dependently induced antiproliferation, caused by apoptosis and autophagy, which was respectively modulated by regulating protein expression levels of Bax and Bcl-2, and LC3.

## 3. Materials and Methods

### 3.1. General Information

All reagents were purchased and used without further purification unless otherwise indicated. Progress of reactions was monitored using TLC visualized by UV lamp (254 nm) or KMnO_4_ developer. Column chromatography was performed using 300 mesh silica gel (Shanxi Nuotai Biological Technology Co., Ltd., Yuncheng, China). Melting points (m.p.) were measured on a Shenguang WRR melting point apparatus (Shanghai Shenguang Instrument Co., Ltd., Shanghai, China). ^1^H- and ^13^C-NMR spectra were recorded using an Agilent 400 MR (Agilent Technology, Santa Clara, CA, USA) in deuterated solvents. Chemicals shifts are reported in parts per million (δ ppm) relative to TMS or the solvent peak. Coupling constants (*J*) are expressed in hertz (Hz). High-resolution mass spectrometry (HRMS) analysis was performed using an Agilent 1290–6545B Q-TOF mass spectrometer (Agilent Technology, Singapore).

### 3.2. Procedure for the Synthesis of 2α-Propargyl Substituted Analogs

#### 3.2.1. Synthesis of Allobetulin (**5**)

Betulin (2.0 g, 4.52 mmol) and *p*-TSA (2.0 g, 11.63 mmol) were added inCH_2_Cl_2_ (100 mL) and refluxed overnight (monitoring by TLC). We removed the solvent under vacuum and the residue was purified by column chromatography on SiO_2_ eluting with CH_2_Cl_2_ to afford compound **5** as a white solid (1.8 g, 4.06 mmol, 89.9%); m.p. 257–258 °C (264–266). ^1^H-NMR (CDCl_3_, 400 MHz) δ: 3.77 (d, *J =* 7.8 Hz, 1H), 3.52 (s, 1H), 3.43 (d, *J =* 7.8 Hz, 1H), 3.19 (dd, *J =* 11.1, 5.1 Hz, 1H), 1.71 (dt, *J =* 13.1, 3.6 Hz, 1H), 0.97 (s, 6H), 0.92 (s, 3H), 0.91 (s, 3H), 0.84 (s, 3H), 0.79 (s, 3H), 0.76 (s, 3H), 0.69 (d, *J =* 9.4 Hz, 1H).^13^C-NMR (CDCl_3_, 100 MHz) δ: 87.9, 78.9, 71.2, 55.5, 51.0, 46.8, 41.4, 40.7, 40.6, 38.9, 38.9, 37.2, 36.7, 36.2, 34.1, 33.9, 32.7, 28.8, 28.0, 27.4, 26.4, 26.4, 26.2, 24.5, 21.0, 18.2, 16.5, 15.7, 15.4, 13.5. HRMS (ESI) calcd for C_30_H_51_O_2_ [M + H]^+^ 443.3889, found 443.3884.

#### 3.2.2. Synthesis of Allobetulon (**6**)

To a solution of allobetulin (1.8 g, 4.06 mmol) in acetone (100 mL) was added freshly prepared Jones’ reagent (18 mL) dropwise at 0 °C, and the solution was stirred for 2 h (monitoring by TLC). The reaction was quenched with MeOH (35 mL) and water (35 mL). The solvent was removed under vacuum, and the aqueous residue was extracted with CH_2_Cl_2_ (3 × 20 mL). We combined the organic layer and dried it with Na_2_SO_4_, then removed the solvent under vacuum to afford compound **6** as a white solid (1.68 g, 3.81 mmol, 93.9%). m.p. 224–226 °C. ^1^H-NMR (CDCl_3_, 400 MHz) δ: 3.78 (d, *J =* 7.8 Hz, 1H), 3.53 (s, 1H), 3.45 (d, *J =* 7.8 Hz, 1H), 2.57–2.36 (m, 2H), 1.94 (ddd, *J =* 12.5, 7.6, 4.6 Hz, 1H), 1.66 (d, *J =* 12.4 Hz, 1H), 1,22 (dd, *J =* 13.3, 4.9 Hz, 1H), 1.08 (s, 3H), 1.03 (s, 3H), 1.01 (s, 3H), 0.94 (s, 3H), 0.93 (s, 3H), 0.92 (s, 6H), 0.79 (s, 3H). ^13^C-NMR (CDCl_3_, 100 MHz) *δ*: 218.2, 87.9, 71.2, 55.0, 50.4, 47.3, 46.8, 41.4, 40.7, 40.5, 39.8, 37.0, 36.7, 36.3, 34.2, 34.1, 33.2, 32.7, 28.8, 26.7, 26.4, 26.4, 26.2, 24.5, 21.5, 21.0, 19.6, 16.3, 15.5, 13.4. HRMS (ESI) calcd for C_30_H_49_O_2_ [M + H]^+^ 441.3733, found 441.3727.

#### 3.2.3. Synthesis of 2α-Propargyl-Allobetulon (**7**)

Compound **6** (1.68 g, 3.81 mmol) was dissolved in DME (80 mL); then 1 M solution of KN(SiMe_3_)_2_ (25 mL, 25 mmol) was added under a nitrogen atmosphere. After 30 min of stirring at room temperature, 1M Et_3_B (27 mL, 27 mmol) in THF was added, and the mixture was stirred for 90 min. Then, a solution of propargyl bromide (2.7 mL, 32 mmol) was added. The reaction mixture was stirred for 6 h under nitrogen (monitoring by TLC), neutralized with 3M HCl (aq), and diluted with water (200 mL). After extraction with EtOAc (3 × 80 mL), the organic layers were combined, washed with saturated NaHCO_3_ and dried over Na_2_SO_4_. The solvent was removed under vacuum and the residue was purified by column chromatography on SiO_2_ via elution with petroleum ether/EtOAc (20/1). Compound **7** was obtained as a white powder (1.18 g, 2.46 mmol, 64.6%). m.p. 184–186 °C. ^1^H-NMR (CDCl_3_, 400 MHz) δ: 3.78 (d, *J =* 7.1 Hz, 1H), 3.53 (s, 1H), 3.45 (d, *J =* 7.8 Hz, 1H), 2.88 (ddt, *J =* 10.0, 8.4, 5.2 Hz, 1H), 2.62 (ddd, *J =* 17.1, 4.4, 2.7 Hz, 1H), 2.37 (dd, *J =* 12.9, 5.6 Hz, 1H), 2.21 (ddd, *J =* 17.1, 8.3, 2.6 Hz, 1H), 1.97 (t, *J =* 2.7 Hz, 1H), 1.15 (s, 3H), 1.07 (s, 3H), 1.06 (s, 3H), 1.04 (s, 3H), 0.94 (s, 3H), 0.91 (s, 3H), 0.80 (s, 3H). ^13^C-NMR (CDCl_3_, 100 MHz) δ: 215.7, 87.9, 83.0, 71.2, 69.4, 57.4, 50.6, 48.3, 46.8, 46.7, 41.4, 41.3, 40.8, 40.7, 37.5, 36.7, 36.2, 34.1, 33.6, 32.7, 28.8, 26.4, 26.3, 26.2, 25.0, 24.5, 21.6, 21.3, 19.5, 19.2, 16.5, 15.8, 13.4. HRMS (ESI) calcd for C_33_H_51_O_2_ [M + H]^+^ 479.3889, found 479.3874.

#### 3.2.4. Synthesis of 2α-Propargyl-Allobetulin (**8**)

Compound **7** (1.18 g, 2.46 mmol) was dissolved in isopropanol (100 mL), NaBH_4_ (186.1 mg, 4.92 mmol) was added, and the mixture was stirred at room temperature overnight (monitoring by TLC). HCl (3M, 40 mL) was dropwise added under 0 °C. The solvent was removed under vacuum, and the residue was extracted with EtOAc (3 × 60 mL); the combined organic layer was washed with saturated NaHCO_3_ and dried over Na_2_SO_4_. We removed the solvent under reduced pressure, and the residue was purified by column chromatography on SiO_2_ eluting with petroleum ether/EtOAc (15/1) to afford compound **8** as a white solid (638.6 mg, 1.33 mmol, 54.1%). m.p. 230–232 °C. ^1^H-NMR (CDCl_3_, 400 MHz) δ: 3.77 (dd, *J =* 7.8, 1.6 Hz, 1H), 3.53 (s, 1H), 3.44 (d, *J =* 7.8 Hz, 1H), 3.03 (dd, *J =* 10.6, 6.3 Hz, 1H), 2.46–2.31 (m, 2H), 2.01 (t, *J =* 2.7 Hz, 1H), 1.86 (dd, *J =* 12.8, 3.8 Hz), 1.83–1.73 (m, 1H), 1.69–1.61 (m, 1H), 1.16–1.06 (m, 1H), 0.99 (s, 3H), 0.98 (s, 3H), 0.93 (s, 3H), 0.92 (s, 3H), 0.89 (s, 3H), 0.80 (s, 3H), 0.70 (s, 3H). ^13^C-NMR (CDCl_3_, 100 MHz) δ: 87.9, 83.0, 81.4, 71.3, 70.0, 55.5, 51.0, 46.8, 44.9, 41.5, 40.8, 40.6, 39.1, 37.4, 36.7, 36.3, 34.8, 34.1, 33.8, 32.7, 28.8, 28.3, 26.4, 26.3, 24.5, 22.3, 21.0, 18.4, 17.3, 16.2, 15.7, 13.5. HRMS (ESI) calcd for C_33_H_53_O_2_ [M + H]^+^ 481.4046, found 481.4038.

### 3.3. General Procedure for Click Reactions

#### 3.3.1. Method A

First, 200 µL of freshly prepared CuSO_4_ solution (1 M) and copper powder (0.1 mmol) were added into a solution of the alkyne (0.30 mmol) and azide (0.20 mmol) in 15 mL ethanol. The resulting mixture was stirred at 45 °C for 48 h until the conversion of azide was completed (monitoring by TLC). The solvent was removed under reduced pressure, and the crude residue was purified by column chromatography on SiO_2_ (5–25% MeOH in CH_2_Cl_2_).

#### 3.3.2. Method B

Azide (0.2 mmol) and alkyne (0.30 mmol) were dissolved in 15 mL *t*-BuOH/H_2_O (1:1, *v*:*v*); then DIPEA (80 μL, 0.48 mmol) was added and stirred for 20 min at 45 °C under nitrogen protection. A solution of CuI (50 mg, 0.26 mmol) in CH_3_CN (1 mL) was added, and the resulting mixture was stirred at 45 °C for 48 h until conversion of azide was completed (monitoring by TLC). The solvent was removed under reduced pressure, and the crude residue was purified by column chromatography on SiO_2_ (6–20% MeOH in CH_2_Cl_2_).

#### 3.3.3. Method C

Azide (0.2 mmol) and alkyne (0.30 mmol) were dissolved in 15 mL *t*-BuOH/H_2_O (1:1, *v*:*v*); then 400 µL fresh prepared sodium ascorbate solution (1 M, 0.4 mmol) and 200 μL CuSO_4_ solution (1M, 0.2 mmol) were added in. The resulting mixture was stirred at 40 °C for 48 h until the conversion of azide was completed (monitoring by TLC). The solvent was removed under reduced pressure, and the crude residue was purified by column chromatography on SiO_2_ (12–15% MeOH in CH_2_Cl_2_).

### 3.4. Procedure for the Preparation of Compounds ***9******a***–***10******i***

#### 3.4.1. 2α-{1*N*[1-(2-deoxy-2β-fluoro-β-d-arabinopentafuranosyl)cytosine-4-yl]-1*H*-1,2,3-triazole-4-yl}-allobetulon (**9a**)

Method B, yield 54.9%, m.p.: decomposition at 200 °C.^1^H-NMR (MeOH-*d_4_*, 400 MHz) δ: 7.93 (d, *J =* 7.4 Hz, 1H), 7.88 (s, 1H), 6.81 (dd, *J =* 12.1, 4.4 Hz, 1H), 5.99 (brs, 1H), 5.34 (dt, *J =* 54.0, 4.6 Hz, 1H), 4.77 (dd, *J =* 20.4, 4.2 Hz, 1H), 4.33 (d, *J =* 12.6 Hz, 1H), 4.24 (d, *J =* 12.2 Hz, 1H), 3.79 (d, *J =* 7.7 Hz, 1H), 3.55 (s, 1H), 3.47 (d, *J =* 7.8 Hz, 1H), 3.26–3.10 (m, 2H), 2.63 (dd, *J =* 14.3, 6.8 Hz, 1H), 2.08 (dd, *J =* 13.0, 5.0 Hz, 1H), 1.15 (s, 3H), 1.08 (s, 3H), 1.07 (s, 3H), 1.05 (s, 3H), 0.95 (s, 3H), 0.91 (s, 3H), 0.82 (s, 3H). ^13^C-NMR (MeOH-*d_4_*, 100 MHz) δ: 218.7, 167.9, 157.9, 146.7, 143.4, 124.1, 99.0 (d, *J =* 6.6 Hz), 96.4 (d, *J =* 193.2 Hz), 89.7, 86.7 (d, *J =* 15.7 Hz), 76.4 (d, *J =* 25.4 Hz), 72.3, 63.4, 59.0, 52.0, 49.6, 48.2, 48.1, 43.6, 42.7, 42.1, 42.0, 38.8, 37.7, 37.3, 35.7, 34.9, 33.9, 29.3, 27.6, 27.5, 27.2, 27.0, 25.7, 24.9, 22.5, 22.1, 20.4, 16.9, 16.4, 13.9.HRMS (ESI) calcd for C_42_H_62_FN_6_O_6_ [M + H]^+^ 765.4715, found 765.4702, calcd forC_42_H_61_FN_6_O_6_Na [M + Na]^+^ 787.4534, found 787.4521.

#### 3.4.2. 2α-{1*N*[1-(2-deoxy-2β-fluoro-β-d-arabinopentafuranosyl)uracil-4-yl]-1*H*-1,2,3-triazole-4-yl}-allobetulon (**9b**)

Method C; yield 52.0%; m.p.: decomposition at 200 °C; ^1^H-NMR (MeOH-*d_4_*, 400 MHz) δ: 7.91 (dd, *J* = 8.2, 1.1 Hz, 1H), 7.87 (s, 1H), 6.79 (dd, *J =* 10.5, 5.5 Hz, 1H), 5.76 (d, *J* = 8.1 Hz, 1H), 5.38 (dt, *J* = 54.2, 5.2 Hz, 1H), 4.84 (dd, *J* = 22.2, 4.9 Hz, 1H), 4.34–4.22 (m, 2H), 3.79 (d, *J* = 7.8 Hz, 1H), 3.55 (s, 1H), 3.48 (d, *J* = 7.8 Hz, 1H), 3.26–3.10 (m, 2H), 2.64 (dd, *J* = 14.4, 7.0 Hz, 1H), 2.08 (dd, *J* = 12.9, 5.2 Hz, 1H), 1.15 (s, 3H), 1.08 (s, 3H), 1.07 (s, 3H), 1.05 (s, 3H), 0.95 (s, 3H), 0.91 (s, 3H), 0.82 (s, 3H). ^13^C-NMR (MeOH-*d_4_*, 100MHz) δ: 218.6, 165.9, 152.0, 146.8, 143.1, 124.0, 102.8, 98.4 (d, *J* = 8.8 Hz), 96.2 (d, *J* = 193.7 Hz), 89.7, 85.4 (d, *J* = 16.9 Hz), 76.3 (d, *J* = 24.8 Hz), 72.3, 63.0, 59.0, 52.0, 49.6, 48.3, 48.1, 43.5, 42.7, 42.1, 42.0, 38.8, 37.7, 37.3, 35.7, 34.9, 33.9, 29.3, 27.6, 27.5, 27.2, 27.0, 25.7, 24.9, 22.5, 22.1, 20.4, 16.9, 16.4, 13.9.HRMS (ESI) calcd for C_42_H_61_FN_5_O_7_ [M + H]^+^ 766.4555, found 766.4539, calcd forC_42_H_60_FN_5_O_7_Na[M + Na]^+^ 788.4374, found 788.4359.

#### 3.4.3. 2α-{1*N*[1-(2-deoxy-2α-fluoro-β-d-ribopentafuranosyl)cytosine-4-yl]-1*H*-1,2,3-triazole-4-yl}-allobetulon (**9c**)

Method B; yield: 45.0%; m.p.: decomposition at 200 °C; ^1^H-NMR (MeOH-*d_4_*, 400 MHz) δ: 8.35 (d, *J* = 7.9 Hz, 1H), 7.93 (s, 1H), 6.46 (dd, *J* = 17.4, 1.4 Hz, 1H), 6.15 (d, *J* = 7.9 Hz, 1H), 5.36 (ddd, *J* = 53.3, 5.2, 1.6 Hz, 1H), 4.97 (dd, *J* = 20.5, 5.2 Hz, 1H), 4.33 (d, *J* = 12.2 Hz, 1H), 4.06 (d, *J* = 12.1 Hz, 1H), 3.79 (d, *J* = 7.8 Hz, 1H), 3.54 (s, 1H), 3.48 (d, *J* = 7.9 Hz, 1H), 3.25–3.08 (m, 2H), 2.71–2.56 (m, 1H), 2.09 (dd, *J* = 12.7, 5.0 Hz, 1H), 1.15 (s, 3H), 1.08 (s, 3H), 1.07 (s, 3H), 1.06 (s, 3H), 0.95 (s, 3H), 0.91 (s, 3H), 0.82 (s, 3H). ^13^C-NMR (MeOH-*d_4_*, 100 MHz) δ: 218.7, 161.8, 148.8, 147.1, 146.7, 124.1, 100.7, 95.4, 93.5 (d, *J* = 191.1 Hz, 1H), 92.9 (d, *J* = 35.5 Hz), 89.7, 72.3, 71.8 (d, *J* = 16.2 Hz, 1H), 64.3, 59.0, 52.0, 49.6, 48.2, 48.1, 43.5, 42.7, 42.1, 42.0, 38.8, 37.6, 37.3, 35.7, 34.9, 33.8, 29.3, 27.6, 27.5, 27.2, 27.0, 25.7, 24.9, 22.5, 22.1, 20.4, 16.9, 16.4, 13.9.HRMS (ESI) calcd for C_42_H_61_FN_6_O_6_Na [M + Na]^+^ 787.4534, found 787.4521.

#### 3.4.4. 2α-{1*N*[1-(2-deoxy-2α-fluoro-β-d-ribopentafuranosyl)uracil-4-yl]-1*H*-1,2,3-triazole-4-yl}-allobetulon (**9d**)

Method C; yield: 48.0%; m.p.: decomposition at 200 °C; ^1^H-NMR (MeOH-*d_4_*, 400 MHz) δ: 7.94 (d, *J* = 8.1 Hz, 1H), 7.92 (s, 1H), 6.41 (dd, *J* = 18.6, 2.4 Hz, 1H), 5.75 (d, *J* = 8.1 Hz, 1H), 5.38 (ddd, *J* = 53.4, 5.3, 2.5 Hz, 1H), 5.00 (dd, *J* = 18.8, 5.3 Hz, 1H), 4.28 (d, *J* = 12.2 Hz, 1H), 4.05 (d, *J* = 12.2 Hz, 1H), 3.79 (d, *J* = 7.9 Hz, 1H), 3.55 (s, 1H), 3.48 (d, *J* = 7.8 Hz, 1H), 3.25–3.09 (m, 2H), 2.70–2.56 (m, 1H), 2.09 (dd, *J* = 12.9, 5.2 Hz, 1H), 1.14 (s, 3H), 1.08 (s, 3H), 1.07 (s, 3H), 1.06 (s, 3H), 0.95 (s, 3H), 0.91 (s, 3H), 0.82 (s, 3H). ^13^C-NMR (MeOH-*d_4_*, 100 MHz) δ: 218.7, 166.1, 152.0, 146.5, 143.8, 124.1, 103.4, 100.4, 93.4 (d, *J* = 189.6 Hz), 92.7 (d, *J* = 35.7 Hz), 89.7, 72.3 (d, *J* = 15.7 Hz), 72.3, 64.8, 59.0, 52.0, 49.6, 48.2, 48.1, 43.5, 42.7, 42.1, 42.0, 38.8, 37.6, 37.3, 35.7, 34.9, 33.9, 29.3, 27.6, 27.5, 27.2, 27.0, 25.7, 24.9, 22.5, 22.1, 20.4, 16.9, 16.4, 14.0.HRMS (ESI) calcd for C_42_H_61_FN_5_O_7_ [M + H]^+^ 766.4555, found 766.4545, calcd forC_42_H_60_FN_5_O_7_Na[M + Na]^+^ 788.4374, found 788.4362.

#### 3.4.5. 2α-{1*N*[1-(β-d-ribopentafuranosyl)cytosine-4-yl]-1*H*-1,2,3-triazole-4-yl}-allobetulon (**9e**)

Method A; yield: 67.9%; m.p.: decomposition at 200 °C; ^1^H-NMR (MeOH-*d_4_*, 400 MHz) δ: 8.00 (d, *J* = 7.4 Hz, 1H), 7.95 (s, 1H), 6.32 (d, *J* = 4.9 Hz, 1H), 5.98 (brs, 1H), 4.68–4.53 (m, 2H), 4.43 (d, *J* = 11.9 Hz, 1H), 3.98 (d, *J* = 11.9 Hz, 1H), 3.79 (d, *J* = 7.9 Hz, 1H), 3.55 (s, 1H), 3.47 (d, *J* = 7.8 Hz, 1H), 3.29–3.07 (m, 2H), 2.60 (dd, *J* = 14.2, 7.1 Hz, 1H), 2.10 (dd, *J* = 12.9, 5.1 Hz, 1H), 1.14 (s, 3H), 1.08 (s, 3H), 1.06 (s, 3H), 1.06 (s, 3H), 0.95 (s, 3H), 0.91 (s, 3H), 0.82 (s, 3H). ^13^C-NMR (MeOH-*d_4_*, 100 MHz) δ: 218.6, 167.8, 158.6, 146.1, 143.7, 124.2, 101.0, 97.0, 93.3, 89.7, 74.3, 73.9, 72.3, 65.9, 59.0, 52.0, 49.6, 48.3, 48.1, 43.6, 42.7, 42.1, 42.0, 38.8, 37.7, 37.3, 35.7, 34.9, 33.9, 29.3, 27.6, 27.5, 27.2, 27.1, 25.7, 24.9, 22.6, 22.1, 20.4, 16.9, 16.4, 14.0.HRMS (ESI) calcd for C_42_H_62_N_6_O_7_Na [M + Na]^+^ 785.4578, found 785.4566.

#### 3.4.6. 2α-{1*N*[1-(β-d-ribopentafuranosyl)uracil-4-yl]-1*H*-1,2,3-triazole-4-yl}-allobetulon (**9f**)

Method A; yield: 70.0%; m.p.: decomposition at 200 °C; ^1^H-NMR (MeOH-*d_4_*, 400 MHz) δ: 8.03 (d, *J* = 8.1 Hz, 1H), 7.90 (s, 1H), 6.33 (d, *J* = 5.1 Hz, 1H), 5.79 (d, *J* = 8.1 Hz, 1H), 4.65–4.56 (m, 2H), 4.44 (d, *J* = 11.9 Hz, 1H), 3.96 (d, *J* = 11.9 Hz, 1H), 3.79 (d, *J* = 7.8 Hz, 1H), 3.55 (s, 1H), 3.47 (d, *J* = 7.8 Hz, 1H), 3.27–3.09 (m, 2H), 2.61 (dd, *J* = 14.3, 6.9 Hz, 1H), 2.10 (dd, *J* = 13.0, 5.2 Hz, 1H), 1.14 (s, 3H), 1.08 (s, 3H), 1.06 (s, 3H), 1.06 (s, 3H), 0.95 (s, 3H), 0.91 (s, 3H), 0.82 (s, 3H). ^13^C-NMR (MeOH-*d_4_*, 100 MHz) δ: 218.6, 166.0, 152.6, 146.2, 142.8, 124.1, 103.7, 101.0, 91.3, 89.7, 74.5, 74.0, 72.3, 66.0, 59.0, 52.0, 49.6, 48.3, 48.1, 43.5, 42.7, 42.1, 42.0, 38.8, 37.6, 37.3, 35.7, 34.9, 33.8, 29.4, 27.6, 27.5, 27.2, 27.1, 25.7, 25.0, 22.6, 22.1, 20.4, 16.9, 16.4, 14.0.HRMS (ESI) calcd for C_42_H_62_N_5_O_8_ [M + H]^+^ 764.4598, found 764.4580, calcd forC_42_H_61_N_5_O_8_Na [M + Na]^+^ 786.4418, found 786.4405.

#### 3.4.7. 2α-{1*N*[1-(2-deoxy-β-d-ribopentafuranosyl)cytosine-4-yl]-1*H*-1,2,3-triazole-4-yl}-allobetulon (**9g**)

Method B; yield: 63.0%; m.p.: decomposition at 200 °C; ^1^H-NMR (DMSO-*d_6_*, 400 MHz) δ: 7.86 (d, *J* = 7.2 Hz, 1H), 7.81 (s, 1H), 7.24 (brs, 2H), 6.60 (t, *J* = 5.0 Hz, 1H), 5.85 (br, 1H), 5.62 (t, *J* = 5.8 Hz, 1H), 5.49 (d, *J* = 5.3 Hz, 1H), 4.68 (m, 1H), 4.21 (dd, *J* = 12.1, 5.7 Hz, 1H), 3.95 (dd, *J* = 12.0, 5.9 Hz, 1H), 3.63 (d, *J* = 7.8 Hz, 1H), 3.40 (s, 1H), 3.34 (d, *J* = 7.8 Hz, 1H), 3.16 (m, 1H), 3.07 (dd,*J* = 14.9, 4.5 Hz, 1H), 2.48 (dd, *J* = 14.9, 7.9 Hz, 1H), 2.20–2.35 (m, 2H), 2.01 (dd, *J* = 12.8, 5.4 Hz, 1H), 1.05 (s, 3H), 1.02 (s, 3H), 0.99 (s, 3H), 0.96 (s, 3H), 0.88 (s, 3H), 0.84 (s, 3H), 0.76 (s, 3H). ^13^C-NMR (DMSO-*d_6_*, 100 MHz) δ: 215.7, 165.7, 154.7, 144.0, 141.5, 121.9, 109.5, 99.3, 86.7, 86.1, 70.5, 70.2, 62.3, 56.7, 49.7, 47.7, 46.2, 46.1, 41.2, 40,8, 40.3, 40.1, 37.7, 37.0, 36.0, 35.9, 33.7, 33.1, 32.4, 28.7, 25.9, 25.8, 25.8, 25.1, 24.2, 21.3, 20.7, 18.7, 15.9, 15.4, 13.2.HRMS (ESI) calcd for C_42_H_62_N_6_O_6_Na [M + Na]^+^ 769.4629, found 769.4611.

#### 3.4.8. 2α-{1*N*[1-(2-deoxy-β-d-ribopentafuranosyl)uracil-4-yl]-1*H*-1,2,3-triazole-4-yl}-allobetulon (**9h**)

Method C; yield: 59.0%; m.p.: decomposition at 200 °C; ^1^H-NMR (MeOH-*d_4_*, 400 MHz) δ: 7.99 (d, *J* = 8.1 Hz, 1H), 7.83 (s, 1H), 6.68 (dd, *J* = 7.2, 5.2 Hz, 1H), 5.75 (d, *J* = 8.1 Hz, 1H), 4.86 (m, 1H), 4.37 (d,*J* = 12.1 Hz, 1H), 4.09 (d, *J* = 12.1 Hz, 1H), 3.79 (d, *J* = 7.9 Hz, 1H), 3.55 (s, 1H), 3.47 (d, *J* = 7.8 Hz, 1H), 3.25–3.09 (m, 2H), 2.68–2.52 (m, 2H), 2.43 (dt, *J* = 13.8, 7.0 Hz, 1H), 2.08 (dd, *J* = 12.9, 5.3 Hz, 1H), 1.14 (s, 3H), 1.08 (s, 3H), 1.07 (s, 3H), 1.06 (s, 3H), 0.95 (s, 3H), 0.91 (s, 3H), 0.82 (s, 3H). ^13^C-NMR (MeOH-*d_4_*, 100 MHz) δ: 218.7, 166.2, 152.2, 146.4, 143.1, 123.9, 103.2, 101.7, 89.7, 87.9, 73.1, 72.3, 64.6, 59.0, 52.0, 48.2, 48.1, 43.6, 42.7, 42.1, 42.0, 38.9, 38.8, 37.7, 37.3, 35.7, 34.9, 33.9, 29.3, 27.6, 27.5, 27.2, 27.1, 25.7, 24.9, 22.6, 22.1, 20.4, 16.9, 16.4, 14.0.HRMS (ESI) calcd for C_42_H_62_N_5_O_7_ [M + H]^+^ 748.4649, found 748.4635, calcd forC_42_H_61_N_5_O_7_Na [M + Na]^+^ 770.4469, found 770.4454.

#### 3.4.9. 2α-{1*N*[1-(2,3-dideoxy-β-d-ribopentafuranosyl)thymine-3-yl]-1*H*-1,2,3-triazole-4-yl}-allobetulon (**9i**)

Method A; yield: 78.0%; m.p.: 193–195 °C; ^1^H-NMR (MeOH-*d_4_*, 400 MHz) δ: 7.92 (d, *J* = 1.2 Hz, 1H), 7.87 (brs, 1H), 6.47 (t, *J* = 6.4 Hz, 1H), 5.40 (dt, *J* = 8.5, 5.5 Hz, 1H), 4.34 (dt, *J* = 5.6, 3.0 Hz, 1H), 3.90 (dd, *J* = 12.2, 3.0 Hz, 1H), 3.78 (d, *J* = 6.5 Hz, 1H), 3.77 (dd, *J* = 15.4, 3.1 Hz, 1H), 3.55 (s, 1H), 3.47 (d, *J* = 7.9 Hz, 1H), 3.27–3.18 (m, 1H), 3.13 (dd, *J* = 14.4, 5.0 Hz, 1H), 2.97–2.84 (m, 1H), 2.72 (ddd, *J* = 14.2, 8.5, 6.3 Hz, 1H), 2.60 (dd, *J* = 14.2, 5.8 Hz, 1H), 2.11 (dd, *J* = 12.9, 5.1 Hz, 1H), 1.90 (d, *J* = 1.1 Hz, 3H), 1.16 (s, 3H), 1.07 (s, 3H), 1.06 (s, 3H), 1.04 (s, 3H), 0.94 (s, 3H), 0.91 (s, 3H), 0.82 (s, 3H). ^13^C-NMR (MeOH-*d_4_*, 100 MHz) δ: 218.3, 166.4, 152.3, 147.9, 146.2, 138.3, 124.1, 111.7, 89.6, 86.7, 86.5, 72.2, 62.2, 60.9, 59.0, 52.0, 49.6, 48.7, 48.1, 43.4, 42.7, 42.1, 42.0, 39.1, 38.9, 37.7, 37.3, 35.7, 34.9, 33.9, 29.4, 27.6, 27.5, 27.2, 25.7, 25.0, 22.6, 22.1, 20.4, 17.0, 16.5, 14.0, 12.7.HRMS (ESI) calcd for C_44_H_64_N_5_O_6_ [M + H]^+^ 746.4857, found 746.4846, calcd forC_44_H_63_N_5_O_6_Na [M + Na]^+^ 768.4676, found 768.4665.

#### 3.4.10. 2α-{1*N*[1-(2-deoxy-2β-fluoro-β-d-arabinopentafuranosyl)cytosine-4-yl]-1*H*-1,2,3-triazole-4-yl}-allobetulin (**10a**)

Method B; yield: 58.7%; m.p.: decomposition at 200 °C; ^1^H-NMR (DMSO-*d_6_*, 400 MHz) δ: 7.90 (s, 1H), 7.77 (d, *J* = 7.4 Hz, 1H), 7.33 (brs, 1H), 7.30 (brs, 1H), 6.76 (dd, *J* = 7.3, 5.6 Hz, 1H), 6.23 (d, *J* = 5.0 Hz, 1H), 5.85 (t, *J* = 5.6 Hz, 1H), 5.80 (d, *J* = 7.3 Hz, 1H), 5.32 (dt, *J* = 55.3, 5.6 Hz, 1H), 4.72 (dt, *J* = 25.0, 4.6 Hz, 1H), 4.58 (d, *J* = 6.4 Hz, 1H), 4.23–4.06 (m, 2H), 3.62 (d, *J* = 7.5 Hz, 1H), 3.39 (s, 1H), 3.33 (d, *J* = 7.5 Hz, 1H), 3.16 (d, *J* = 12.8 Hz, 1H), 2.71 (dd, *J* = 10.2, 6.5 Hz, 1H), 2.29 (dd, *J* = 14.4, 9.8 Hz, 1H), 1.90–1.75 (m, 1H), 1.63 (d, *J* = 11.7 Hz, 1H), 0.91 (s, 3H), 0.89 (s, 3H), 0.87 (s, 3H), 0.84 (s, 3H), 0.75 (s, 3H), 0.75 (s, 3H), 0.72 (s, 3H), 0.52 (t, *J* = 12.8 Hz, 1H). ^13^C-NMR (DMSO-*d_6_*, 100 MHz) δ: 165.6, 154.7, 144.9, 141.8, 122.1, 95.5 (d, *J* = 10.2 Hz), 94.7 (d, *J* = 191.4 Hz), 94.2, 86.7, 83.0, 80.4, 74.2 (d, *J* = 24.8 Hz), 70.2, 60.9, 55.2, 50.4, 46.1, 44.4, 40.9, 40.2, 40.1, 39.0, 36.8, 36.0, 35.9, 35.4, 33.7, 33.3, 32.4, 28.8, 28.5, 28.5, 25.9, 25.9, 25.8, 24.2, 20.5, 18.1, 16.9, 16.6, 15.4, 13.3.HRMS (ESI) calcd for C_42_H_64_FN_6_O_6_ [M + H]^+^ 767.4871, found 767.4860.

#### 3.4.11. 2α-{1*N*[1-(2-deoxy-2β-fluoro-β-d-arabinopentafuranosyl)uracil-4-yl]-1*H*-1,2,3-triazole-4-yl}-allobetulin (**10b**)

Method C; yield: 59.9%; m.p.: decomposition at 200 °C; ^1^H-NMR (MeOH-*d_4_*, 400 MHz) δ: 7.92 (d, *J* = 8.3 Hz, 1H), 7.89 (s, 1H), 6.81 (dd, *J* = 10.2, 5.5 Hz, 1H), 5.76 (d, *J* = 8.1 Hz, 1H), 5.41 (dt, *J* = 54.5, 5.3 Hz, 1H), 4.86 (dd, *J* = 22.4, 5.0 Hz, 1H), 4.30 (s, 2H), 3.77 (d, *J* = 7.8 Hz, 1H), 3.54 (s, 1H), 3.46 (d, *J* = 7.8 Hz, 1H), 3.24–3.11 (m, 1H), 2.83 (d, *J* = 10.7 Hz, 1H), 2.54 (dd, *J* = 14.2, 9.0 Hz, 1H), 2.06–1.88 (m, 1H), 1.70 (dd, *J* = 12.9, 2.8 Hz, 1H), 0.98 (s, 6H), 0.93 (s, 3H), 0.91 (s, 3H), 0.83 (s, 3H), 0.82 (s, 3H), 0.81 (s, 3H), 0.74 (d, *J* = 9.3 Hz, 1H), 0.62 (t, *J* = 12.6 Hz, 1H). ^13^C-NMR (MeOH-*d_4_*, 100 MHz) δ: 165.9, 152.0, 147.0, 143.1, 123.8, 102.8, 98.3 (d, *J* = 8.6 Hz), 96.22 (d, *J* = 193.9 Hz), 89.6, 85.4 (d, *J* = 16.7 Hz), 83.1, 76.2 (d, *J* = 25.1 Hz, 1H), 72.3, 62.9, 57.2, 52.4, 48.1, 46.3, 42.7, 41.9, 41.9, 40.5, 38.5, 37.7, 37.3, 37.2, 35.7, 35.1, 33.9, 29.8, 29.4, 29.1, 27.6, 27.6, 27.2, 25.0, 22.3, 19.7, 17.8, 17.1, 16.4, 14.1.HRMS (ESI) calcd for C_42_H_63_FN_5_O_7_ [M + H]^+^ 768.4712, found 768.4700.

#### 3.4.12. 2α-{1*N*[1-(2-deoxy-2α-fluoro-β-d-ribopentafuranosyl)cytosine-4-yl]-1*H*-1,2,3-triazole-4-yl}-allobetulin (**10c**)

Method B; yield: 50.9%; m.p.: decomposition at 200 °C; ^1^H-NMR (MeOH-*d_4_*, 400 MHz) δ: 7.96 (s, 1H), 7.95 (d, *J* = 7.4 Hz, 1H), 6.38 (dd, *J* = 19.0, 1.7 Hz, 1H), 5.94 (d, *J* = 7.6 Hz, 1H), 5.32 (ddd, *J* = 53.9, 5.1, 2.0 Hz, 1H), 5.01 (dd, *J* = 19.7, 5.3 Hz, 1H), 4.29 (d, *J* = 12.2 Hz, 1H), 4.10 (d, *J* = 12.1 Hz, 1H), 3.77 (d, *J* = 7.9 Hz, 1H), 3.54 (s, 1H), 3.46 (d, *J* = 7.8 Hz, 1H), 3.18 (dd, *J* = 14.4, 3.1 Hz, 1H), 2.83 (d, *J* = 10.8 Hz, 1H), 2.54 (dd, *J* = 14.4, 8.9 Hz, 1H), 2.02–1.88 (m, 1H), 1.72, (dd, *J* = 13.1, 3.3 Hz, 1H), 0.99 (s, 6H), 0.94 (s, 3H), 0.91 (s, 3H), 0.84 (s, 3H), 0.82 (s, 3H), 0.81 (s, 3H), 0.74 (d, *J* = 9.5 Hz, 1H), 0.63 (t, *J* = 12.7 Hz, 1H). ^13^C-NMR (MeOH-*d_4_*, 100 MHz) δ: 168.2, 157.9, 146.7, 144.5, 124.0, 100.4, 96.8, 94.12 (d, *J* = 35.1 Hz), 93.78 (d, *J* = 189.4 Hz), 89.7, 83.1, 72.4 (d, *J* = 17.5 Hz), 72.3, 65.0, 57.2, 52.5, 48.2, 46.3, 42.8, 42.0, 41.9, 40.5, 38.5, 37.7, 37.3, 37.3, 35.7, 35.1, 33.9, 29.8, 29.3, 29.1, 27.6, 27.6, 27.2, 25.0, 22.3, 19.7, 17.8, 17.1, 16.3, 14.0.HRMS (ESI) calcd for C_42_H_63_FN_6_O_6_Na [M + Na]^+^ 789.4691, found 789.4676.

#### 3.4.13. 2α-{1*N*[1-(2-deoxy-2α-fluoro-β-d-ribopentafuranosyl)uracil-4-yl]-1*H*-1,2,3-triazole-4-yl}-allobetulin (**10d**)

Method C; yield: 54.0%; m.p.: decomposition at 200 °C; ^1^H-NMR (MeOH-*d_4_*, 400 MHz) δ: 7.95 (s, 1H), 7.94 (d, *J* = 8.2 Hz, 1H), 6.41 (dd, *J* = 18.6, 2.0 Hz, 1H), 5.75 (d, *J* = 8.1 Hz, 1H), 5.39 (ddd, *J* = 53.4, 5.1, 2.3 Hz, 1H), 5.01 (dd, *J* = 18.7, 5.3 Hz, 1H), 4.29 (d, *J* = 12.2 Hz, 1H), 4.07 (d, *J* = 12.2 Hz, 1H), 3.78 (d, *J* = 7.8 Hz, 1H), 3.53 (s, 1H), 3.46 (d, *J* = 7.7 hz, 1H), 3.19 (dd, *J* = 14.1, 2.7 Hz, 1H), 2.83 (d, *J* = 10.8 Hz, 1H), 2.53 (dd, *J* = 14.5, 9.1 Hz, 1H), 2.02–1.86 (m, 1H), 1.72 (dd, *J* = 13.0, 2.9 Hz, 1H), 0.99 (s, 6H), 0.94 (s, 3H), 0.91 (s, 3H), 0.84 (s, 3H), 0.82 (s, 3H), 0.81 (s, 3H), 0.75 (d, *J* = 9.2 Hz, 1H), 0.63 (t, *J* = 12.5 Hz, 1H). ^13^C-NMR (MeOH-*d_4_*, 100 MHz) δ: 166.0, 152.0, 146.8, 143.8, 124.0, 103.5, 100.4, 93.4 (d, *J* = 190.1 Hz), 92.7 (d, *J* = 35.4 Hz), 89.7, 83.1, 72.3, 72.3 (d, *J* = 16.2 Hz), 64.9, 57.2, 52.5, 48.2, 46.3, 42.8, 42.0, 41.9, 40.5, 38.6, 37.7, 37.4, 37.3, 35.7, 35.1, 33.9, 29.8, 29.3, 29.1, 27.6, 27.6, 27.2, 25.0, 22.3, 19.7, 17.8, 17.1, 16.3, 14.0.HRMS (ESI) calcd for C_42_H_63_FN_5_O_7_ [M + H]^+^ 768.4712, found 768.4702.

#### 3.4.14. 2α-{1*N*[1-(β-d-ribopentafuranosyl)cytosine-4-yl]-1*H*-1,2,3-triazole-4-yl}-allobetulin (**10e**)

Method B; yield: 56.9%; m.p.: decomposition at 200 °C; ^1^H-NMR (DMSO-*d_6_*, 400 MHz) δ: 7.82 (d, *J* = 7.5 Hz, 1H), 7.79 (s, 1H), 7.33 (brs, 1H), 7.29 (brs, 1H), 6.22 (d, *J* = 5.7 Hz, 1H), 5.81 (d, *J* = 7.4 Hz, 1H), 5.69 (t, *J* = 5.9 Hz, 1H), 5.42 (d, *J* = 8.4 Hz, 1H), 5.40 (d, *J* = 9.3 Hz, 1H), 4.54 (d, *J* = 6.7 Hz, 1H), 4.44–4.36 (m, 2H), 4.25 (dd, *J* = 11.8, 6.3 Hz, 1H), 3.83 (dd, *J* = 11.8, 5.5 Hz, 1H), 3.62 (d, *J* = 7.9 Hz, 1H), 3.38 (s, 1H), 3.31 (d, *J* = 7.9 Hz, 1H), 3.21–3.11 (m, 1H), 2,71 (dd,*J* = 10.5, 6.9 Hz, 1H), 2.26 (dd, *J* = 14.4, 9.8 Hz, 1H), 1.88–1.74 (m, 1H), 1.66 (dd, *J* = 12.9, 2.6 Hz, 1H), 0.91 (s, 3H), 0.89 (s, 3H), 0.87 (s, 3H), 0.84 (s, 3H), 0.76 (s, 3H), 0.74 (s, 3H), 0.71 (s, 3H), 0.54 (t, *J* = 12.4 Hz, 1H). ^13^C-NMR (DMSO-*d_6_*, 100 MHz) δ: 165.6, 155.3, 144.4, 142.0, 121.9, 98.4, 94.8, 90.0, 86.7, 80.5, 72.5, 71.7, 70.2, 64.1, 55.2, 50.3, 46.1, 44.5, 40.8, 40.2, 40.1, 39.0, 36.7, 36.0, 35.8, 35.7, 33.7, 33.3, 32.4, 28.7, 28.6, 28.5, 25.9, 25.8, 25.8, 24.2, 20.5, 18.1, 16.9, 16.6, 15.4, 13.3.HRMS (ESI) calcd for C_42_H_64_N_6_O_7_Na [M + Na]^+^ 787.4734, found 787.4716.

#### 3.4.15. 2α-{1*N*[1-(β-d-ribopentafuranosyl)uracil-4-yl]-1*H*-1,2,3-triazole-4-yl}-allobetulin (**10f**)

Method A; yield: 52.2%; m.p.: decomposition at 200 °C; ^1^H-NMR (MeOH-*d_4_*, 400 MHz) δ: 8.03 (d, *J* = 8.1 Hz, 1H), 7.93 (s, 1H), 6.36 (d,*J* = 5.5 Hz, 1H), 5.79 (d, *J* = 8.1 Hz, 1H), 4.66–4.56 (m, 2H), 4.46 (d, *J* = 11.9 Hz, 1H), 3.98 (d, *J* = 11.9 Hz, 1H), 3.77 (d, *J* = 7.8 Hz, 1H), 3.54 (s, 1H), 3.46 (d, *J* = 7.8 Hz, 1H), 3.24–3.12 (m, 1H), 2.83 (d, *J* = 10.7 Hz, 1H), 2.52 (dd, *J* = 14.2, 9.1 Hz, 1H), 2.05–1.87 (m, 1H), 1.79–1.67 (m, 1H), 0.99 (s, 6H), 0.94 (s, 3H), 0.91 (s, 3H), 0.84 (s, 3H), 0.82 (s, 3H), 0.81 (s, 3H), 0.74 (d, *J* = 9.4 Hz, 1H), 0.64 (t, *J* = 12.7 Hz, 1H). ^13^C-NMR (MeOH-*d_4_*, 100MHz) δ: 166.0, 152.6, 142.8, 103.7, 101.1, 91.3, 89.7, 83.0, 74.6, 74.0, 72.3, 65.9, 57.2, 52.4, 48.1, 46.4, 42.7, 41.9, 41.9, 40.6, 38.5, 37.7, 37.3, 37.3, 35.7, 35.1, 33.9, 29.9, 29.4, 29.1, 27.6, 27.6, 27.2, 25.0, 22.3, 19.7, 17.9, 17.2, 16.3, 14.1.HRMS (ESI) calcd for C_42_H_64_N_5_O_8_ [M + H]^+^ 766.4755, found 766.4740.

#### 3.4.16. 2α-{1*N*[1-(2-deoxy-β-d-ribopentafuranosyl)cytosine-4-yl]-1*H*-1,2,3-triazole-4-yl}-allobetulin (**10g**)

Method B; yield: 60.1%; m.p.: decomposition at 200 °C; ^1^H-NMR (MeOH-*d_4_*, 400 MHz) δ: 8.05 (d, *J* = 7.5 Hz, 1H), 7.87 (s, 1H), 6.69 (t, *J* = 5.9 Hz, 1H), 5.95 (brs, 1H), 4.84 (t, *J* = 6.8 Hz, 1H), 4.40 (d, *J* = 12.0 Hz, 1H), 4.12 (d, *J* = 12.1 Hz, 1H), 3.77 (d, *J* = 7.9 Hz, 1H), 3.54 (s, 1H), 3.46 (d, *J* = 7.8 Hz, 1H), 3.17 (dd, *J* = 14.1, 2.5 Hz, 1H), 2.83 (d, *J* = 10.8 Hz, 1H), 2.60–2.39 (m, 3H), 2.05–1.89 (m, 1H), 1.71 (dd, *J* = 13.0, 2.9 Hz, 1H), 0.99 (s, 6H), 0.94 (s, 3H), 0.91 (s, 3H), 0.84 (s, 3H), 0.82 (s, 3H), 0.81 (s, 3H), 0.74 (d, *J* = 9.4 Hz, 1H), 0.63 (t, *J* = 12.7 Hz, 1H). ^13^C-NMR (MeOH-*d_4_*, 100 MHz) δ: 167.8, 158.1, 146.6, 143.1, 123.8, 101.8, 96.6, 89.7, 89.0, 83.1, 73.0, 72.3, 64.6, 57.2, 52.5, 48.2, 46.3, 42.8, 42.0, 41.9, 40.5, 39.6, 38.6, 37.7, 37.3, 37.3, 35.7, 35.1, 33.9, 29.8, 29.3, 29.1, 27.6, 27.6, 27.2, 24.9, 22.3, 19.7, 17.8, 17.1, 16.3, 14.0.HRMS (ESI) calcd for C_42_H_64_N_6_O_6_Na [M + Na]^+^ 771.4785, found 771.4767.

#### 3.4.17. 2α-{1*N*[1-(2-deoxy-β-d-ribopentafuranosyl)uracil-4-yl]-1*H*-1,2,3-triazole-4-yl}-allobetulin (**10h**)

Method C; yield: 63.0%; m.p.: decomposition at 200 °C; ^1^H-NMR (MeOH-*d_4_*, 400 MHz)δ: 7.99 (d, *J* = 8.1 Hz, 1H), 7.85 (s, 1H), 6.70 (dd, *J* = 7.1, 5.2 Hz, 1H), 5.75 (d, *J* = 8.1 Hz, 1H), 4.85 (m, 1H), 4.38 (d, *J* = 12.1 Hz, 1H), 4.11 (d, *J* = 12.1 Hz, 1H), 3.78 (d, *J* = 7.8 Hz, 1H), 3.54 (s, 1H), 3.46 (d, *J* = 7.8 Hz, 1H), 3.17 (dd, *J* = 14.3, 3.1 Hz, 1H), 2.83 (d, *J* = 10.8 Hz, 1H), 2.67–2.49 (m, 2H), 2.45 (dt,*J* = 13.8, 7.0 Hz, 1H), 2.03–1.87 (m, 1H), 1.71 (dd, *J* = 13.0, 3.3 Hz, 1H), 0.99 (s, 3H), 0.98 (s, 3H), 0.94 (s, 3H), 0.91 (s, 3H), 0.84 (s, 3H), 0.82 (s, 3H), 0.81 (s, 3H), 0.74 (d, *J* = 9.1 Hz, 1H), 0.63 (t, *J* = 12.6 Hz, 1H). ^13^C-NMR (MeOH-*d_4_*, 100 MHz) δ: 166.3, 152.3, 146.6, 143.1, 123.7, 103.2, 101.6, 89.7, 88.0, 83.1, 73.1, 72.3, 64.6, 57.2, 52.5, 48.2, 46.3, 42.8, 42.0, 41.9, 40.5, 38.9, 38.5, 37.7, 37.3, 37.3, 35.7, 35.1, 33.9, 29.8, 29.3, 29.1, 27.6, 27.6, 27.2, 24.9, 22.3, 19.7, 17.8, 17.1, 16.3, 14.0.HRMS (ESI) calcd for C_42_H_64_N_5_O_7_ [M + H]^+^ 750.4806, found 750.4790.

#### 3.4.18. 2α-{1*N*[1-(2,3-dideoxy-β-d-ribopentafuranosyl)thymine-3-yl]-1*H*-1,2,3-triazole-4-yl}-allobetulin (**10i**)

Method B; yield: 78.2%; m.p.: 192–193 °C; ^1^H-NMR (MeOH-*d_4_*, 400 MHz) δ: 7.91 (s, 1H), 7.87 (s, 1H), 6.48 (t, *J* = 6.4 Hz, 1H), 5.40 (dt, *J* = 8.5, 5.6 Hz, 1H), 4.35 (dt, *J* = 5.7, 3.0 Hz, 1H), 3.91 (dd, *J* = 12.2, 2.9 Hz, 1H), 3.78 (d, *J* = 7.5 Hz, 1H), 3.77 (dd, *J* = 12.3, 3.2 Hz, 1H), 3.54 (s, 1H), 3.47 (d, *J* = 7.8 Hz, 1H), 3.19 (dd,*J* = 14.5, 2.6 Hz, 1H), 2.93 (dt, *J* = 12.5, 6.4 Hz, 1H), 2.82 (d, *J* = 10.8 Hz, 1H), 2.74 (ddd, *J* = 14.3, 8.5, 6.2 Hz, 1H), 2.51 (dd, *J* = 14.5, 9.2 Hz, 1H), 1.91 (s, 3H), 1.70 (dd, *J* = 13.1, 3.3 Hz, 1H), 0.99 (s, 3H), 0.99 (s, 3H), 0.94 (s, 3H), 0.91 (s, 3H), 0.84 (s, 3H), 0.82 (s, 3H), 0.81 (s, 3H), 0.75 (d, *J* = 9.1 Hz, 1H), 0.65 (t, *J* = 12.7 Hz, 1H). ^13^C-NMR (MeOH-*d_4_*, 100 MHz) δ: 166.5, 152.4, 148.2, 138.4, 123.8, 111.8, 89.7, 86.8, 86.5, 83.1, 72.3, 62.2, 60.9, 57.2, 52.5, 48.2, 46.5, 42.8, 42.0, 41.9, 40.5, 39.1, 38.5, 37.7, 37.4, 37.3, 35.7, 35.1, 33.9, 29.9, 29.3, 29.1, 27.6, 27.6, 27.2, 24.9, 22.4, 19.7, 17.8, 17.1, 16.3, 14.0, 12.6.HRMS (ESI) calcd for C_43_H_66_N_5_O_6_ [M + H]^+^ 748.5013, found 748.5004, calcd for C_43_H_65_N_5_O_6_Na [M + Na]^+^ 770.4833, found 770.4814.

### 3.5. X-ray Structure of Compound ***9******c***

Colorless, block-like, single crystals of compound **7c** were obtained after recrystallization from CH_2_Cl_2_ and methanol. A crystal of dimensions 0.15 × 0.1 × 0.09 mm was selected to collect a room temperature (293K) X-ray crystallographic dataset. The data were collected on a Gemini E diffractometer (Agilent Technology, Oxyford, UK) with graphite monochromated Cu Kα radiation (λ = 1.54184 Å).

### 3.6. Cell Culture

HepG2, MNK-45, MCF-7, SW620, and A549 cell lines were purchased from Procell Life Science & Technology Co., Ltd. (Wuhan, China). SMMC-7721 cell line was purchased from BeNa Culture Collection (Beijing, China). MNK-45, SMMC-7721, and SW620 cells were cultured in Roswell Park Memorial Institute (RPMI) 1640 medium (Solarbio, Beijing, China); andHepG2, MCF-7,and A549 cells were cultured in Dulbecco’s modified eagle’s medium (DMEM) with 10% fetal bovine serum (FBS) and 1% penicillin/streptomycin (Solarbio, Beijing, China).All cells were incubated at 37 °C in a humidified 5% CO_2_ atmosphere.

### 3.7. Cell Viability Assay

Cell viability was measured by the CCK-8 assay. Confluent cells in the well-state were cultured in 96-well plates (5–10 × 10^4^ cells/mL). After cells were attached to the plate, compounds with various concentrations were applied at 37 °C for 48 h. Then, the medium containing drugs was replaced with 10% CCK-8 solution prepared by using the fresh serum-free medium. After incubation at 37 °C for 30 min, the medium was transferred to the 96-well plates and measured at 450 nm using an enzyme-linked immunosorbent assay (ELISA) reader at 450 nm.

### 3.8. Flow Cytometry Assay

Flow cytometry analysis was applied for apoptosis detection. Firstly, SMMC-7721 cells were adjusted to 2 × 10^5^/mL, inoculated into a six-well plate, and placed in an incubator at 37 °C containing 5% CO_2_ saturated humidity overnight. After the cells were fully attached to the plate; 1, 5, 10, and 15 μM of **8d** were administrated to the cells for 48 h. Cells were collected and stained with Annexin V-FITC and PI. Subsequently, flow cytometry was used for detection.

Flow cytometry analysis was applied for cell cycle detection. Firstly, SMMC-7721 cells were adjusted to 2 × 10^5^/mL, inoculated into a six-well plate, and placed in an incubator at 37 °C containing 5% CO_2_ in saturated humidity overnight. After the cells were fully attached to the plate, 1, 5, 10, and 15 μM of **8d** were administrated to the cells for 24 h. After cells were collected and fixed with 70% ethanol, PI was applied to stain the cells; subsequently, onboard testing by flow cytometry (Cytoflex S (Beckman Coulter, Brea, CA, USA)) was conducted.

### 3.9. Western Blot Analysis

Cells were treated with different concentrations of **10d** for 48 h, and then were harvested, and total protein was extracted using lysis buffer (Solarbio, Beijing, China). Equal lysates were separated by sodium dodecyl sulfate-polyacrylamide gel electrophoresis, and then transferred to PVDF membranes (Solarbio, Beijing, China). Subsequently, the membranes were blocked with 5% nonfat milk in TBST (50 mM Tris-HCl (pH 7.4), 150 mM NaCl and 0.1% Tween 20) for 2 h and incubated with the following primary antibodies at 4 °C overnight: LC3 (Protrintech Group, Wuhan, China), Bcl-2, GAPDH, and Bax (SAB, Beijing, China). In sequence, the membranes were washed and probed with goat anti-rabbit IgG/HRP (Beijing Biosynthesis Biotechnology Co., Ltd.) at room temperature for 2 h. The signals were detected by ECL Plus Hypersensitive luminescence solution (Solarbio, Beijing, China) and an ECL system (Beijing Oriental Science and Technology Development Co., Ltd., Beijing, China). The quantitative analysis of mean pixel density was performed by the ImageJ^®^ software.

### 3.10. Statistical Analysis

All experiments were performed at least three times, and statistical analysis was performed by using Microsoft Excel. Data were presented as mean ± SD, and statistical significance was determined by ANOVA with the post hoc test. The *p*-value < 0.05 indicated a statistically significant difference.

## 4. Conclusions

The new series of allobetulon/allobetulin–nucleoside conjugates (**9****a**–**10i**) were synthesized, and their antitumor activities were evaluated. Among them, compounds **9b**, **9e**, **10a**, and **10d** showed promising antiproliferative activity in six tested cell lines, compared to zidovudine, cisplatin, and oxaliplatin. Regarding the structure–activity relationship, introducing nucleosides to the scaffolds (**7** and **8**) can improve their potency. However, their potency did not significant correspond to their substituted types of nucleotide base. Based on their antiproliferative activity, compound **10d** can be considered a promising candidate for further investigation. We investigated the potential mechanism for compound **10d**. Compound **10d** dose-dependently induced cell apoptosis and autophagy in SMMC cells, resulting in antiproliferation and G0/G1 cell cycle arrest by regulating protein expression levels of Bax, Bcl-2, and LC3. Consequently, the nucleoside-conjugated allobetulin (**10d**) evidenced that nucleoside substitution is an available strategy for improving allobetulon/allobetulin antitumor activity based on our present study.

## Data Availability

The data presented in this work are available in the article and Appendix A.

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
