# Peer review of "Synthesis and Biological Evaluation of Novel Allobetulon/Allobetulin–Nucleoside Conjugates as AntitumorAgents"

_molecules, 2022, doi:10.3390/molecules27154738_

Round 1

Reviewer 1 Report

The manuscript in its present form cannot be published. It needs to be corrected by an English expert. The Authors should decide if they want to write it in the Present or Past Tense and do not mix both tenses.

Some examples of mistakes in the text:

Introduction                                                                                                        line 44 - remove "been"                                                                             

line 54 - verb is missing after to

line 57 - should be "is obtained in the rearangement.."

lines 65-75 - make no sence, should be re-written by an English exoert.

Biological evaluation                                                                                         line 126 except not excepting?

line 135 the sentence starting with "Particularly makes no sence

line 139 should be "was" not "and"

These are just few examples, the manuscript needs much more English editting.

As to the biological part, the Authors tested the cytotoxicity of their new compounds on 6 cell lines. One selected analog was further investigated on one cell line. It was shown that this analog induced apoptosis and  influenced cell cycle. That is really not much. All cytotoxic compounds usually induce apoptosis. Conclusions are very short. Nothing is written about apoptosis pathway. What about cell cycle, at which phase was it stopped? What is a possible mechanism of action of the selected compound?

Author Response

The manuscript in its present form cannot be published. It needs to be corrected by an English expert. The Authors should decide if they want to write it in the Present or Past Tense and do not mix both tenses.

Some examples of mistakes in the text:

Introduction

line 44 - remove "been"                                                                             

line 54 - verb is missing after to

line 57 - should be "is obtained in the rearangement.."

lines 65-75 - make no sence, should be re-written by an English exoert.

Biological evaluation

line 126 except not excepting?

line 135 the sentence starting with "Particularly makes no sence

line 139 should be "was" not "and"

These are just few examples, the manuscript needs much more English editting.

Response: Thanks for your suggestions. We re-edited and polished the manuscript.

As to the biological part, the Authors tested the cytotoxicity of their new compounds on 6 cell lines. One selected analog was further investigated on one cell line. It was shown that this analog induced apoptosis and influenced cell cycle. That is really not much. All cytotoxic compounds usually induce apoptosis. Conclusions are very short. Nothing is written about apoptosis pathway. What about cell cycle, at which phase was it stopped? What is a possible mechanism of action of the selected compound?

Response: As you suggested, a comprehensive investigation is important to understand the mechanism of action for compounds/drugs. However, our current study focused on designing and synthesizing a series of target compounds to figure out the potential candidates or leads. After obtaining these derivatives and testing their potency, we took a further step to understand its possible mechanism of action. In our primary study for compound 10d, Cell cycle analysis showed an increased accumulation of cells in the G0/G1 phase after treatment with compound 10d (Figure 3B). Therefore, induction of G0/G1 cell cycle arrest in SMMC-7721 cell lines implied compound 10d reduced cell proliferation by induction of G0/G1 cell cycle arrest. Some factors can trigger the G0/G1 cell cycle arrest, including apoptosis, Cyclin-dependent kinase inhibition, the regulation of tumor suppressors, and autophagy. To clarify the potential factors, we firstlyinvestigated the critical regulators of cell apoptosis by western blot analysis. Pro-apoptotic protein Bax and the anti-apoptotic protein Bcl-2 were well-known factors linked to the regulation of apoptosis. SMMC-7721 cells were treated with 10d (1, 5, 10, 15 μM) for 48 h, and we then examined the associated protein levels (Bax, Bcl-2, and GAPDH) by Western blotting. As shown in Figure 3E, the expression of the pro-apoptotic protein Bax upregulated, and the anti-apoptotic protein Bcl-2 significantly downregulated in a dose-dependent manner. Based on these results, compound 10d might lead to G0/G1 phase arrest in SMMC-7721 cells through apoptosis by Bax and Bcl-1 regulation. Additionally, G0/G1 phase arrest may be induced by autophagy. Therefore, the autophagy marker, LC3, was examined by Western blotting. Cells were treated with 10d (1, 5, 10, 15 μM) for 48 h, and then measured the expression of LC3. As shown in Figure 3E, compound 10d significantly induced the LC3 expression. Taken together, compound 10d dose-dependently induced antiproliferation, caused by apoptosis and autophagy, which was respectively modulated by regulating protein expression levels of Bax and Bcl-2, as well as LC3. 

Reviewer 2 Report

This manuscript describes the synthesis and the biological evaluation of some novel allobetulin/allubetulon-nucleoside hybrid derivatives. The novel conjugated compounds were tested for their antiproliferative activity, which was found to be promising for some of them. Although this work and its results are quite interesting, the manuscript should undergo major revision before it could be considered for publication.

- A thorough text editing is required.

-The novel derivatives are allubetulon/allubetulin-nucleoside conjugates (not ‘conjugations’). This should be corrected in the title and throughout the text.

- The numbering of the allobetulin and allobetulon conjugates must be different from the parent compounds (page 3, scheme 2). Thus, derivatives of allubetulon should be numbered as 8a-i, allobetulin should be numbered as 9 and derivatives of allubetulin should be numbered as 10a-i, throughout the main text and the SI file.

- The synthesis of compounds 5 (Allobetulin) and 6 (Allobetulon) has been previously described under the same experimental conditions, thus the corresponding references should be provided by the authors, both in the main text and in the experimental section, as well.

- The α-orientation of the propargyl group is not unambiguously proved by the NOESY spectrum of compound 8 (figure S11). Indeed, the structure of compound 8 is complicated and a number of peaks appear between 0.8-1.0 ppm in the spectrum. The overlapping of the peaks of H-24 and H-5 may lead to false conclusions. In order to avoid this, I suggest that the authors should also provide the COSY spectrum of compound 8, together with a second, more focused, picture of the NOESY spectrum (ideally, a zoom picture of the area between 0-2.0 ppm).

 - The purity of the target derivatives seems to be very high, according to the NMR spectra figures provided in the SI. Nevertheless, the authors should also make a statement concerning their purity and the way it was checked, in the general information of the experimental section. Usually, a purity 95% is necessary for the compounds that undergo pharmacological evaluation.

- The authors should explain why they used zidovudine in the section 2.2.1.

- The authors should provide some information about the statistical analysis of the results presented in section 2.2.2 and figure 3.

- A graphical abstract was not included in the files sent to the reviewers.

Author Response

Comments and Suggestions for Authors

 This manuscript describes the synthesis and the biological evaluation of some novel allobetulin/allubetulon-nucleoside hybrid derivatives. The novel conjugated compounds were tested for their antiproliferative activity, which was found to be promising for some of them. Although this work and its results are quite interesting, the manuscript should undergo major revision before it could be considered for publication.

- A thorough text editing is required.

Response: Thanks for your suggestions. We reedited and polished the manuscript.

-The novel derivatives are allubetulon/allubetulin-nucleoside conjugates (not ‘conjugations’). This should be corrected in the title and throughout the text.

Response: Thank you for your careful review. We corrected the mistakes.

- The numbering of the allobetulin and allobetulon conjugates must be different from the parent compounds (page 3, scheme 2). Thus, derivatives of allubetulon should be numbered as 8a-i, allobetulin should be numbered as 9 and derivatives of allubetulin should be numbered as 10a-i, throughout the main text and the SI file.

Response: Thank you for your advice. We corrected the mistakes.

- The synthesis of compounds 5 (Allobetulin) and 6 (Allobetulon) has been previously described under the same experimental conditions, thus the corresponding references should be provided by the authors, both in the main text and in the experimental section, as well.

Response: Thank you for your advice. We added the corresponding references in the manuscript.

- The α-orientation of the propargyl group is not unambiguously proved by the NOESY spectrum of compound 8 (figure S11). Indeed, the structure of compound 8 is complicated and a number of peaks appear between 0.8-1.0 ppm in the spectrum. The overlapping of the peaks of H-24 and H-5 may lead to false conclusions. In order to avoid this, I suggest that the authors should also provide the COSY spectrum of compound 8, together with a second, more focused, picture of the NOESY spectrum (ideally, a zoom picture of the area between 0-2.0 ppm).

Response: Thank you for your advice. We provided the COSY spectrum and the zoomed NOESY spectrum of compound 8 in supporting information. Furthermore, theα-orientation of the propargyl group is confirmed by the X-ray crystallographic structure of compound 9c, the NOESY spectrum is another evidence.

 - The purity of the target derivatives seems to be very high, according to the NMR spectra figures provided in the SI. Nevertheless, the authors should also make a statement concerning their purity and the way it was checked, in the general information of the experimental section. Usually, a purity ≥95% is necessary for the compounds that undergo pharmacological evaluation.

Response: Thank you for your valuable advice. We added that in the manuscript.

- The authors should explain why they used zidovudine in the section 2.2.1.

Response: Thank you for your review. We used zidovudine as positive reference in order toconfirm the antitumor activity of the conjugates. Compound 7, 8 and zidovudine have no antitumor activity, but their conjugates (9i and 10i) have potent antitumor activity and indicated the conjugates maybe have a novel antitumor mechanismand we will investigate it deeply in future.

- The authors should provide some information about the statistical analysis of the results presented in section 2.2.2 and figure 3.

Response: Thank you for your valuable advice. All CCK8 assay data were presented with mean ± standard deviation (mean +- S.D.).  Statistical analysis was performed by using Microsoft Excel.

- A graphical abstract was not included in the files sent to the reviewers.

Response: Thank you for your careful review. We added it.

Reviewer 3 Report

The article written by Yanli Wang et al. presents synthesis and biological evaluation of novel allobetulon/allobetulin-nucleoside conjugations as antitumor agents. The synthesis of drugs that use natural sources is now an important part of medical chemistry research. In the present research, the authors used betulin, a naturally occurring pentacyclic triterpene. The presented research is interesting, but the manuscript may be accepted for publication after considering the comments below:

1. The reported products are optically active. Are the products enantiomerically pure? Please, measure the optical rotation for each compound.

2. The authors should also investigate the toxicity of the compounds against the healthy cell line.

3. Are the obtained derivatives more active than the starting betulin? Please, add betulin as a positive control.

Author Response

Comments and Suggestions for Authors

 The article written by Yanli Wang et al. presents synthesis and biological evaluation of novel allobetulon/allobetulin-nucleoside conjugations as antitumor agents. The synthesis of drugs that use natural sources is now an important part of medical chemistry research. In the present research, the authors used betulin, a naturally occurring pentacyclic triterpene. The presented research is interesting, but the manuscript may be accepted for publication after considering the comments below:

  1. The reported products are optically active. Are the products enantiomerically pure? Please, measure the optical rotation for each compound.

Response: Thank you for your advice. Except C-2 and C-3, the chiral properties of other carbons didn’t change. The α-orientation of the propargyl group of C-2 was confirmed by the X-ray crystallographic structure of compound 9c and NOESY. The β-orientation of the hydroxylgroup of C-3 was confirmed by NOESY and coupling constant between H-3 and H-2(3JH(2),H(3)= 10.6 Hz, CDCl3). The products are enantiomerically pure.

  1. The authors should also investigate the toxicity of the compounds against the healthy cell line.

Response: Thank you for your advice. We will test the toxicity of the compounds against the healthy cell line in further study.

  1. Are the obtained derivatives more active than the starting betulin? Please, add betulin as a positive control.

Response: We tested the IC50 of betulin for five cell lines and the results are listed below. The target compounds were more active than betulin. In fact, betulin only have moderate antitumor activity.

Compd.

IC50 (μM)

SMMC-7721

HepG2

MNK-45

SW620

MCF-7

A549

Betulin

82.9

>100

55.05

83.70

30.6

87.39

Round 2

Reviewer 1 Report

The manuscript has been improved. The last semtence in the abstract should be changed. This compound will probably never become a drug.

Author Response

Thank you for your advice. We have revised the content in abstract and conclusion. Indeed, this series of nucleotide-conjugates has a far way to go if they apply to clinical use. However, we evidenced that nucleoside-substitution is an available strategy to improve allobetulon/allobetulin’s antitumor activity based on our present study.

Reviewer 2 Report

The authors have made all the necessary corrections, thus the manuscript may now be considered for publication.

Author Response

We appreciate your advice that improves our manuscript a lot.